# Observing of the super-Planckian near-field thermal radiation between graphene sheets

Jiang Yang[1], Wei Du[1], Yishu Su[1], Yang Fu[1], Shaoxiang Gong[1], Sailing He[1] & Yungui Ma [1]

Thermal radiation can be substantially enhanced in the near-field scenario due to the tunneling of evanescent waves. Monolayer graphene could play a vital role in this process owing to its strong infrared plasmonic response, however, which still lacks an experimental verification due to the technical challenges. Here, we manage to make a direct measurement about plasmon-mediated thermal radiation between two macroscopic graphene sheets using a custom-made setup. Super-Planckian radiation with efficiency 4.5 times larger than the blackbody limit is observed at a 430-nm vacuum gap on insulating silicon hosting substrates. The positive role of graphene plasmons is further confirmed on conductive silicon substrates which have strong infrared loss and thermal emittance. Based on these, a thermophotovoltaic cell made of the graphene–silicon heterostructure is lastly discussed. The current work validates the classic thermodynamical theory in treating graphene and also paves a way to pursue the application of near-field thermal management.

[1] State Key Lab of Modern Optical Instrumentation, Centre for Optical and Electromagnetic Research, College of Optical Science and Engineering, Zhejiang University, Hangzhou 310058, China. Correspondence and requests for materials should be addressed to Y.M. (email: yungui@zju.edu.cn)

Objects with temperature will radiate the infrared light arising from the mechanical oscillation of inner charges and the emitted light spectrum can be statistically characterized by the classic Plank's radiation law. For the far-field scenario, a blackbody that absorbs all the impinging light has the capacity to yield the maximum energy transfer efficiency compared with nature materials. When a receiver is placed at a distance far smaller than the thermal de Broglie wavelength ($\lambda_{th} = hc/K_bT$, $h$-Planck constant, $c$-light velocity in vacuum, $K_b$-Boltzmann constant, and $T$-temperature), the thermally excited evanescent waves that carry high density of states of photons can tunnel through the subwavelength gaps and thus substantially enhance the near-field heat transfer efficiency[1,2]. For polar dielectrics ($SiO_2$, $Al_2O_3$, h-BN, etc.), this enhancement effect will be more obvious due to the excitation and participation of bounded phononic surface polaritons that will strengthen the electromagnetic coupling across the gaps[3]. Thermal fluctuating electrodynamic (TFE) theory has been developed to characterize the near-field heat transfer process, initially with analytical formulas describing regular shaped objects (e.g., semi-infinite surface[4] or sphere pair[5]) and more recently with sophisticated numerical schemes such as fluctuating surface current formalism[6] or quasi normal mode analysis[7] that are able to deal with objects with more complex geometries. The continuum theory of fluctuating electromagnetics, as recently predicted by Chiloyan et al.[8], will fail to describe the heat transfer over sub-nanometer gaps where direct phonon tunneling will replace thermal radiation to dominate the heat conduction. Various applications based on the framework of near-field heat transfer have been proposed such as thermophotovoltaic (TPV) cell[9], thermal scanning imaging[10], thermal condenser or rectifier[11], etc. On the other hand, the experiment on the near-field thermal radiation has been long delayed due to the technical difficulties primarily in controlling the gap distances although the first measurement was reported in 1970[12]. The super-blackbody radiation was only experimentally observed till 2008 by Hu et al. between two glass plates at a fixed gap distance of 1600 nm[13]. In recent years, there has been increased attentions and various advanced technologies have been developed to verify the conception of super-Planckian radiation such as between metals[14–16], doped semiconductors[17,18], polar dielectrics[3,15,16], or metamaterials[19], with fixed or variable gaps in the configuration of either microsphere (or point)-plate[3,15,17–21] or (micro-) plate–plate[14,16]. Quite recently, Cui et al. has pushed the gap limit down to sub-nanometers using the state-of-the-art measurement apparatuses[22], while Bernardi et al. has successfully characterized the near-field heat flux between two macroscopic silicon wafers at nanoscale gaps[23].

So far, the abnormal thermal radiation effect between usual metals or dielectrics has been well examined and the underlying description formula have been verified in the experiment. Graphene (Gr) as the single-carbon atomic layer material has unique plasmon polariton response in the infrared window which provides a promising material option to manipulate the excitation and emission of thermal photons. Compared with highly doped plasmonic semiconductors, graphene has the outstanding advantages including tunable Fermi level with a linear Dirac band, for example, by voltage-gating, easy transfer to or be integrated with different substrates and good thermal stability. Theoretically, enhancement of the heat transfer efficiency by graphene[24] and their applications such as in TPV[9,25], thermal circuits[26] or other thermal management devices[27] have been explored in the literature. Far-field thermal radiation of graphene sheets[28] or nanostructures[29] heated in high temperatures have been experimentally inspected before. However, there is no experiment to directly examine the role of Gr plasmons in mediating the near-field heat transfer except that Van Zwol et al.

ever tried to inspect this effect between a $SiO_2$ microsphere and a graphene sheet[30]. Technically, it is not easy to obtain high quality and large-sized graphene samples, in particular for the near-field application that has high requirements on surface morphologies and substrate alignment.

In this work, we managed to have a direct measurement of plasmon-mediated near-field heat transfer between two graphene sheets with nanoscale vacuum gaps. Silicon substrates with well-controlled physical properties has been utilized to host the graphene layer. The heterostructure also provides a platform to modify the charge density of graphene by forming a Schottky junction at the Gr–Si interface[31–33]. Thermal radiation with power density far larger than the Blackbody limit is first observed with intrinsic Si substrates (negligible loss) as assisted by the inter-layer EM coupling. In a comparison, the heat transfer between the bare Si–Si substrates is fractionally small. The measurement is repeated using highly doped silicon substrates that alone can give rise to strong heat transfer in a near-field pair as material loss and evanescent wave tunneling increase. Covering their surfaces with Gr sheets could further enhance the heat transfer efficiency although in this case, graphene will exhibit a lower Fermi level resulted from the electron doping over the junction. The measured results are well reproduced analytically with the input of the measured material properties, thus explicitly validating the classic TFE theory. The current work also provides a useful material structure and measurement platform to explore the full features of Gr plasmons controlled thermal photon manipulations, which is specifically discussed in the end for TPVs[9,24,25].

## Results

**Near-field heat transfer measurement.** The monolayer graphene used in the work is a commercial product made by chemical vapor phase deposition (CVD) (7440-44-0, XFNANO Mater Tech Co., Ltd, China) originally grown on a copper foil. Standard technique has been employed to transfer the macroscopic Gr sheets onto silicon substrates which are adhered together by the Van der Waals force[34]. The sample is cut into 2×2 cm squares for experiment. All these processes including the measurement are conducted in a cleaning environment. The details are addressed in the Methods section. Silicon hosting substrates are employed here primarily for three reasons. First, their controllable infrared properties through doping can be tailored to identify the Gr plasmon's role in the near-field heat transfer. Second, silicon wafer with an ultra-small bow is achievable (in this work, a 500 °C post-annealing in vacuum is carried out to release the cutting-caused wafer bending)[35]. And lastly, the Gr–Si heterostructure may be a potential assembly for the important TPV or photo-thermoelectric applications. They have a roughness about 1–2 nm and a maximum bow value less than 20 nm determined by a Laser interferometer (ZYGO OMP-035/M) with an area of 2×2 cm. Figure 1a sketches our home-made measurement setup for the near-field thermal radiation. The two graphene layers are separated by a vacuum gap (experimentally with a pressure <5 × 10⁻⁶ Torr) assisted by four AZ photoresist (PR) posts (diameter = 50 μm, height = 500 nm, edge distance = 0.15 mm and conductivity = 0.18 W m⁻¹ K⁻¹). The PR posts have good physical stability below 150 °C and their height decides the gap value. In our setup, the top Gr/Si assembly (temperature $T_e$) behaves as the emitter sourced by a same sized heating plate through a copper spreader and the bottom Gr/Si assembly (temperature $T_r$) is the receiver sunk by a temperature electric controller (TEC) (1-12705, Realplay, China) through another copper spreader. Silicone grease is used to improve the boundary thermal contact between the stacks. A 200-nm thick titanium film has been

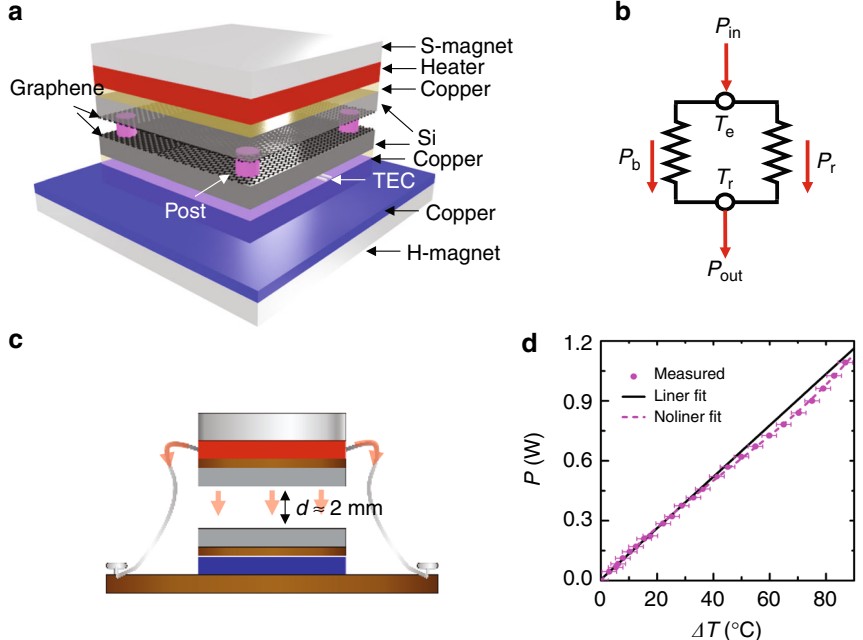

**Fig. 1** Measurement setup. **a** A schematic of the home-made measurement setup. From top to bottom, the emitter side consists of S-magnet, heater, copper spreader, and Si/Gr and the receiver side consisting of Gr/Si, copper spreader, TEC layer, copper spreader, and H-magnet. Four photoresist posts are used to separate the emitter and the receiver. **b** The equivalent thermal circuit of our measurement setup. The input power $P_{in}$ is split into the background branch with power $P_b$ and the near-field radiation with power $P_r$. **c** Schematic of the background heat loss measurement. The receiver is maintained at 30 °C and displaced from the emitter at a macroscopic vacuum distance of 2 mm. The input power (heat) will be dissipated along the metal wires and via radiation. **d** The relation of input power with the temperature increment of the emitter. The experimental data points are denoted by the dots and the solid/dashed line represents the linear/non-linear fitting, from which we obtain the background heat loss $P_b(\Delta T)$. The error bar for the temperature is 2.3 °C, derived from the five times repeating measurement. Figure 1 by Yang J et al.

deposited on the top surface of the silicon substrate in order to block the photons emitted from the grease layer. Two temperature sensors are inserted into the center of the top and bottom copper spreaders through the drilled slots to monitor $T_e$ and $T_r$. A pair of magnets consisting of a 5×5×3 cm³ NdFeB hard magnet (H-magnet, bottom) and a 2×2×1 cm³ iron soft magnet (S-magnet, top) are used to strengthen the contact and mechanical stability of the system by forming a clamping force of about 10 N. The exposed surface of the top half unit is coated by a 200-nm thick aluminum film to minimize the background radiation. In the measurement, we have assumed the Gr/Si assembly has the same temperature with the copper spreader since the contacting thermal resistance is negligibly small compared with the radiation one. The Gr/Si/Cu triplets at the receiver side are thermally maintained at a stable temperature around 30 ± 0.30 °C (equal to the environment temperature $T_b$) assisted by the TEC cell. The maximum temperature difference ($\Delta T = T_e - T_r$) we obtain is ~50 °C at a vacuum gap around 500 nm and ~70 °C as the gap is larger than 1 μm.

As sketched in Fig. 1b, the input electric power $P_{in}$ after subtracted by the resistive heating loss of electrode wires is majorly dissipated through two channels: one we called is the background power loss ($P_b$) including one part conducted along the electrode wires of source and temperature sensors and the other part radiated toward the surrounding space and the second is the net power ($P_r$) radiated via the vacuum gap. As the total input power is known, it is crucial to correctly evaluate the background transfer branch $P_b$, which in our case is done in a separate experiment measuring the temperature rising in a far-field configuration. As schematically shown in Fig. 1c, for the background measurement, the top half of the device is displaced from the bottom one at a gap of ~2 mm supported by the four rigid metal wires (belonging to the source and sensors). The other ends of these wires are screw

fixed with the bottom steel sink platform. By this structure, $P_b$ at different $T_e$ is estimated through a simple relation: $P_b(T_e) = P_{in}(T_e) - P_r^{Far}(T_e)$, where $P_r^{Far} = \varepsilon\sigma(T_e^4 - T_b^4)$ ($\varepsilon$-emissivity and $\sigma$-coefficient) is the power emitted via the Gr/Si surface. As $T_e$ is relatively small, the radiation via other Al-film coated surfaces is neglected, i.e., assuming that $P_b$ is primarily caused by the heat flowing out along the electrode wires. Figure 1d plots the measured temperature rising curve as a function of the input power. The experimental data could be well fitted by a binomial expression $P_{in}(\Delta T) = a \times \Delta T + b \times \Delta T^4$ with the fitting coefficients $a = 0.014$ W K⁻¹ and $b = 5.2 \times 10^{-9}$ W K⁻⁴. Note here we use $\Delta T = T_e - T_e^0$ and $T_e^0 = T_b = 30$ °C. When $\Delta T < 40$ °C, the temperature is linearly dependent on the input power, indicating that the background radiation could be neglected at small temperature changes. Thereafter, in the following, the near-field radiated heat is evaluated by the equation $P_r = P_{in}(\Delta T) - a \times \Delta T - P_{AZ}(\Delta T)$, where $P_{AZ}$ is the heat conducted via the four posts.

**Graphene on intrinsic insulating silicon.** Our first experimental sample utilizes an insulating silicon (i-Si) substrate (resistivity > 20,000 ω cm) without doping so that the substrate contribution to the near-field heat flux could be minimized and the role of the Gr plasmons can be more clearly examined. The maximum temperature change for the emitter is less than 100 °C. The variation of infrared properties due to temperature change for both silicon and graphene can be practically neglected. Figure 2a gives a typical three-dimensional (3D) atomic force microscopy (AFM) morphology picture of the Gr–Si heterostructure in an area of 20×20 μm². Statistical measurement shows the surface roughness of our sample is less than ~50 nm, which is relatively small when considering the multiple physical and chemical processes involved

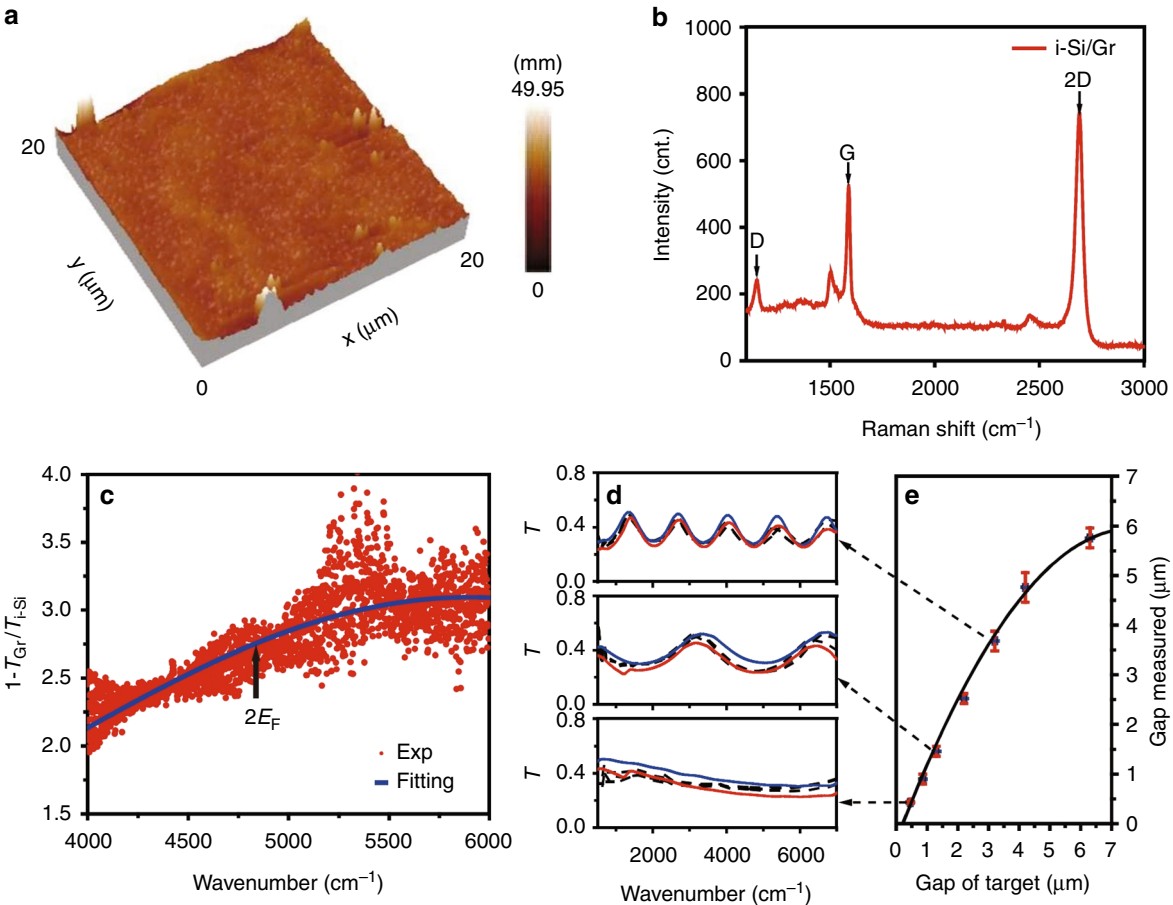

**Fig. 2** Characterization of i-Si/Gr properties. **a** A typical 3D AFM picture of the i-Si/Gr sample. The largest roughness of the sample is about 50 nm. **b** The Raman spectrum of the i-Si/Gr sample. The monolayer feature of graphene is identified by the strong 2D resonance peak. **c** The absorption spectrum of graphene on intrinsic silicon. The red dots are measured results by FTIR and the blue line is a Gaussian function fitting. $T_{Gr}$ and $T_{Si}$ denote the transmittance with and without the graphene cover, respectively. The "$2E_F$ onset" of the inter-band transition is estimated at about 4832.8 cm$^{-1}$, corresponding a Fermi level at $E_F = 0.27$ eV. **d** Measured (dashed lines) and calculated (solid lines) FTIR spectra at three different target gap sizes of 3.7, 1.4, and 0.43 μm. The uncertainty of gap size is considered here so that the calculated lines (red and blue) could cover the experimental curves measured at various spatial points. The statistic value for the smallest gap we designed is about 430 ± 25 nm. **e** Correspondence between the measured and the target gap sizes. The gap uncertainty is shown by the error bar derived from the measurements at multiple points. The black curve is a guidance line for eye, which shows nearly a linear relationship when the gap value is within few micron meters. Figure 2 by Yang J et al.

in the sample preparation. As a consequence, in the experiment, we set the height of the photoresist posts or gap size at values larger than 400 nm. Figure 2b gives the Raman spectrum of the Gr–Si heterostructure. The typical peaks from the D-mode and G-mode resonances are observed at 1149.98 and 1591.46 cm$^{-1}$, respectively. Their small peak intensity ratio indicates the high quality of the CVD graphene[36]. According to the G-mode position, we can estimate the Fermi level of graphene at $E_F = 0.27$ eV due to the electron doping from the absorbed molecules, which is close to the previous reported value (0.32 eV) also for a CVD graphene[37]. In addition, the strong single 2D peak identifies the dominant monolayer nature of our graphene sample[38]. Figure 2c gives the absorption spectrum of the Gr–Si heterostructure measured by a Fourier transform infrared (FTIR) spectrometer, where $T_{Gr}$ and $T_{i-Si}$ denote the transmittance with and without graphene, respectively. There is a broad peak at 4832.8 cm$^{-1}$, associated with doping inhomogeneity, temperature effect and other broadening mechanisms. The spectrum has no obvious change as we raise the temperature from 30 to 70 °C, indicating the thermal stability of the heterostructure. The middle-IR broad absorption curve can be phenomenologically fitted by a Gaussian step function[37]. From the full width at half-maximum (FWHM), we can estimate the

"$2E_F$ onset" of the inter-band transition, as indicated by the arrow in Fig. 2c, which corresponds to a Fermi level of 0.27 ± 0.03 eV, highly consistent with that estimated from the Raman spectrum. The gap distance $d$ of the silicon substrates is mainly decided by the height of the photoresist posts. The real value may have a certain distribution due to the bow of wafer or roughness, etc. As shown in Fig. 2d, we use the measured FTIR spectra (dashed lines) and the theoretical ones (solid lines) to evaluate the gap value. The red and blue lines calculated at different $d$ represent the edges of the experimental lines measured at various spatial points. Figure 2e shows there is a good linear correspondence between the measured and the target gap distances when it is within few micron meters. The uncertainty is about 25 nm at a central distance of 430 nm and gets larger when the distance increases. The measured values have been used in the following numerical calculations for the heat transfer efficiency.

Figure 3a gives the measured heat flux density $h$ varied as a function of temperature difference $\Delta T$ for the samples with and without graphene. The measured gap size is 430 ± 25 nm and the receiver's temperature is controlled at $T_r = 30$ °C. For the bare Si–Si assembly, the measured heat flux density (blue dot) fluctuates below the line (black) representing the Blackbody

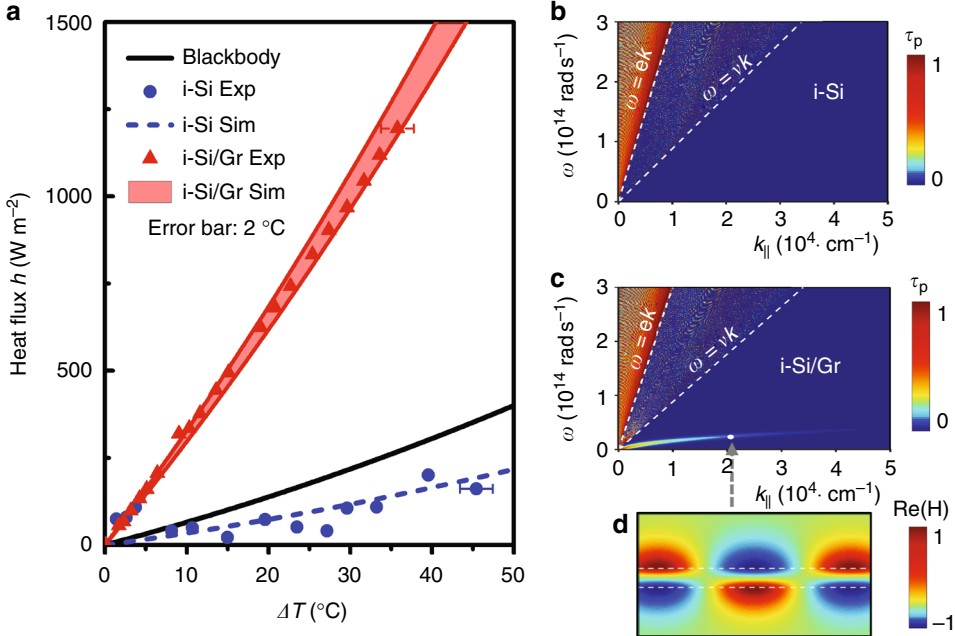

**Fig. 3** Heat flux density and transmission possibility between i-Si/Gr pair. **a** The measured heat flux density (symbols) compared with the analytical predictions (shaded region and dashed line) at different emitter temperature variation between i-Si substrates with or without graphene. The gap distance for both case is 230 nm. The blue dots are measured points for the bare i-Si pair, while the blue dashed line is the numerical prediction calculated using the measured material parameters. The red triangles are measured points between Gr-covered i-Si substrates, while the red shaded region denotes the corresponding theoretical heat flux density considering $E_F = 0.27 \pm 0.03$ eV and $d = 430 \pm 25$ nm. The temperature uncertainty is 2 °C derived from the five times repeating measurement. The black line denotes the flux density of Blackbody radiation. **b**, **c** p-polarization transmission possibility $\tau_p$ ($\omega, k_{||}$) between bare i-Si pair and i-Si/Gr pair, respectively. The interference strip patterns in the transmission map within the free-space light cone is caused by the FP resonance in an ultra-low loss planar stack system. **d** The mode pattern denoted by the magnetic component between the i-Si/Gr pair at ($2.5 \times 10^{13}$ rad s$^{-1}$, $2.3 \times 10^4$ rad cm$^{-1}$) and $d = 500$ nm. Figure 3 by Yang J et al.

radiation. Although the intrinsic silicon has an ultra-small imaginary permittivity ($\sim 10^{-4}$ derived from the FTIR absorption measurement), the thick substrate will still produce certain amount of thermal radiation and this effect will be enlarged in the near-field case with the involvement of evanescent wave tunneling[23]. The heat flux from the naturally oxidized ultrathin SiO$_2$ layer (typically $\sim 1$ nm[39]) on the silicon surface is negligibly small. The blue dashed line in Fig. 3a is the analytically calculated heat flux with the input of measured material parameters, which overall has a good agreement with the experimental data. The heat flux is substantially enhanced when covering the substrate with a graphene layer as shown (red triangles) in Fig. 3a. At $\Delta T = 36$ °C, the heterostructure gives rise to a strong flux density at $h = 1195$ W m$^{-2}$, nearly 4.5 times larger than the Blackbody radiation ($h = 268$ W m$^{-2}$).The total radiation heat flow at this case is about 0.48 W for our sample, which is nearly four times larger than that ($P_{AZ} \sim 0.12$ W) conducted along the four supporting posts. The red shaded region profiles the theoretical heat flux distribution calculated by considering the fluctuations: $d = 430 \pm 25$ nm and $E_F = 0.27 \pm 0.03$ eV. The collision time factor for a CVD graphene used here is $\tau_{Gr} = 100$ fs[40]. The experiment data is generally well reproduced by the theory. Here we may overlook the radiation contribution from the surface residuals but they do have indirect influences by affecting the Fermi level of graphene and their near-field coupling strength.

To have a deeper understanding about the physical picture, Fig. 3b, c plot the calculated p-polarization photon transmission probability $\tau_p$ ($\omega, k_{||}$) across the vacuum gap without and with graphene at $\Delta T = 50$ °C and $d = 500$ nm, respectively[41]. The metallic ground is described by a Drude model with the data input from ref. [42]. Usually, the p-polarized contribution is dominant in the near-field flux[43]. Without graphene, only

traveling waves inside the free-space light cone (from y-axis to the line $\omega = ck$) has a high transmissivity, in particular for those near the edge of the light cone that have longer light paths in the substrate. The inhomogeneous transmission pattern reflects the Fabry–Perot (FP) resonance features of our low loss planar system. The tunneling possibility for evanescent waves inside the silicon light cone (defined by the line $\omega = v/k$ with $v$ the light velocity in silicon) is rather small, which will become obvious only using much thicker substrates or at smaller gaps (<100 nm). When a graphene layer is introduced, as shown in Fig. 3c, the transmission feature for traveling waves has no obvious change, which accords to the nature of graphene that has low far and mid-IR response due to Pauli block[44]. In addition, there appears a new and p-polarized evanescent wave band below the silicon light cone arising from the near-field EM coupling in an asymmetric super-mode pattern (Fig. 3d). These bounded surface modes carry high density of state of photons and can largely raise the evanescent wave tunneling and thus overall thermal radiation efficiency, as found in the measurement. This effect will be more obvious at smaller gaps or replacing Si with a low-index substrate that will enhance the bandwidth and intensity of the bounded modes[45].

**Graphene on doped silicon.** In the second experiment, we will repeat the experiment using doped ($\sim 10^{18}$ cm$^{-3}$) n-type silicon substrates that can strongly modify the dielectric properties of the background and grapheme layers. Especially, the doped silicon (d-Si)/Gr heterostructure is interesting for the potential applications in TPV cells or thermophotoelectric sensors due to the formation of a Schottky junction at their interface[33,46], which will modulate the infrared response of graphene as we observed

before[31]. The dielectric properties of the heterostructure are derived from the measured FTIR spectrum. Further information is given in the Methods section. Figure 4a plots the measured reflectance curves of the d-Si substrate with (shifted upward by 0.05) and without grapheme at different substrate temperatures. Note in this case the substrate is totally opaque in the IR region and has no transmission. The reflectance of both samples has almost no variation when the temperature varies in 30–70 °C, implying the stability of charge densities in both substrate and graphene layer under the agitation of moderate temperature changes. This result is expected for a highly doped semiconductor[47]. Then, a Drude model fitting (solid line in Fig. 4a) is applied here to retrieving the dielectric constant of the substrate. The results are plotted in Fig. 4b. Our doped silicon substrate will have a metallic response below the plasmon frequency of $5.6 \times 10^{14}$ rad s$^{-1}$ (3.3-μm wavelength) accompanied with a dramatic change on both real and imaginary permittivity. Figure 4c gives the real (blue dots) and imaginary (red squares) parts of the spectral sheet conductivity of graphene directly retrieved from the measured reflectance with the parameters of silicon input from Fig. 4b[48]. From terahertz to far-IR, the monolayer graphene will also electromagnetically behave as an optical metal and can be described by a Drude model conductivity resulted from the intraband electron scattering. In Fig. 4c with the shaded regions, the measured conductivity data is well fitted by the Drude model with a slight fluctuation of Fermi level at $E_F = 0.15 \pm 0.03$ eV. The $E_F$ value on the d-Si substrate is nearly half of that (0.27 eV) of the intrinsic silicon, indicating the occurrence of electron transfer from the d-Si substrate to the p-type graphene (a general property for a CVD-grew graphene) assisted by the Schottky heterojunction at the Gr–Si interface[37,49].

Figure 5a plots the measured heat flux density (symbols) between two d-Si substrates with and without graphene at various temperature differences. The receiver side is maintained at $T_r = 30$ °C. For both samples, we design two different gaps at $d = 430$ and 1150 nm. Their measured uncertainties are ±25 nm and ±50 nm, respectively. First, a large heat flux density exceeding the Blackbody radiation is experimentally observed between the bare silicon substrates with a 1150 nm gap, indicating the participation of thermally excited evanescent waves. The heat flux density is substantially strengthened when the gap size is reduced to 430 nm. These results agree with the theoretical calculations quite well by considering the fluctuation of gap. Similar super-Planckian near-field thermal radiation between doped silicon wafers was also recently reported by Watjen et al.[50]. Introducing a graphene

layer will bring no obvious change to the radiation efficiency at a large gap (1150 nm). The increment becomes obvious when the gap is shrunk to 430 nm. At an example point of $\Delta T = 37$ °C, the heat flux density increases about 11% after introducing graphene. The measured results are consistent with the analytical predictions (shaded regions) calculated using the measured material and structural parameters. In addition, it is noted that with the same configurations, the d-Si/Gr heterostructure pair will give a much larger heat flux than the i-Si/Gr pair, primarily due to the loss-induced additional channel for the generation and transmission of thermal photons using doped silicon.

Figure 5b, c gives the p-polarization transmission probability $\tau_p$ ($\omega$, $k_\parallel$) between the d-Si pairs without and with graphene at $\Delta T = 50$ °C and $d = 500$ nm, respectively. For the bare d-Si pair, compared with that shown in Fig. 3b for the i-Si pair, the transmission map becomes homogenous due to the disappearance of FP resonance and the evanescent portion (inside the silicon light cone but outside the free-space light cone) grows largely. The density of thermal source significantly increases in the lossy silicon substrates, which greatly raises the transmission possibility for both traveling and evanescent waves. As a result, it leads to the exceeding Blackbody's heat flux. In addition, we also observe a low-frequency evanescent wave transmission band right below the line $\omega = v/k$, arising from the plasmonic near-field coupling between the silicon (negative $\varepsilon$ in this region) surfaces. A typical asymmetric field pattern for this super-mode is shown in Fig. 5d by magnetic field. With a top graphene layer, this band will be extended in the wavenumber domain. Compared with that on the i-Si substrate, the asymmetrical super-mode nature of the evanescent wave band does not change (Fig. 5e) but the dispersion diagram shifts to lower frequencies and in this case the inter-layer coupling will be weakened in particular at large gaps (e.g., $d > 100$ nm)[45]. On the other hand, the momentum and mode pattern matches will improve the near-field coupling of plasmonic polaritons between graphene and silicon substrate. As a consequence, more evanescent waves can be transferred to strengthen the heat flux, as shown in Fig. 5a.

In the above, the role of graphene plasmon polaritons in enhancing the near-field coupling[51] and heat transfer efficiency has been experimentally observed at different substrates. The results not only validate the continuum theory of fluctuating electromagnetics in dealing with the single layer material, but also provide an indirect evidence about the existence of this highly bounded surface modes and their tunability in our macroscopic characterization. For this aim, it is crucial to have high

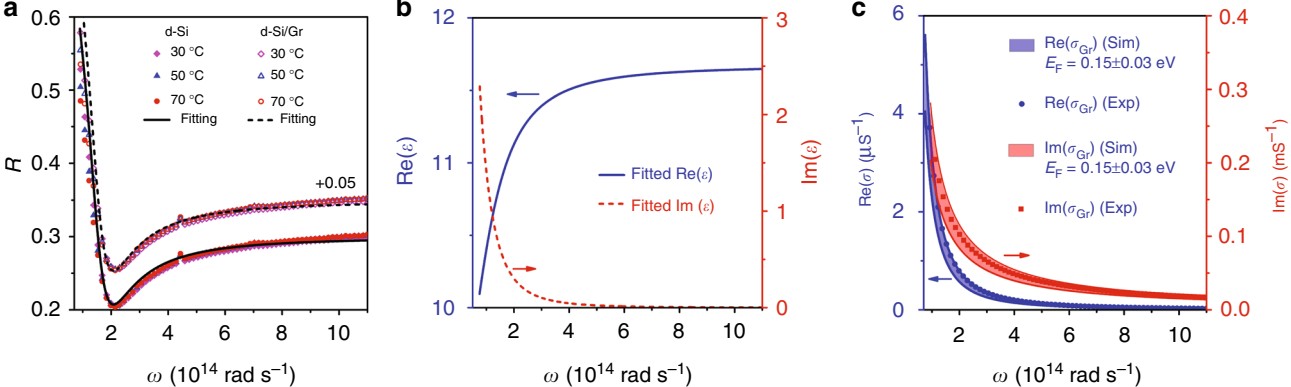

**Fig. 4** Characterization of d-Si/Gr properties. **a** The measured reflection spectra of the d-Si and d-Si/Gr samples at different temperatures. We have intentionally shifted up the value of d-Si/Gr by +0.05 for better vision. The black solid/dashed line is the fitting curve by considering a Drude conductivity model for both Si and graphene in the far-mid-IR range. **b** The retrieved real (blue) and imaginary (red) parts of permittivity for the doped silicon substrate. **c** The retrieved real (blue dots) and imaginary (red squares) parts of sheet conductivity for the graphene layer. The shaded regions represent the theoretical conductivity distribution calculated using $E_F = 0.15 \pm 0.03$ eV. Figure 4 by Yang J et al.

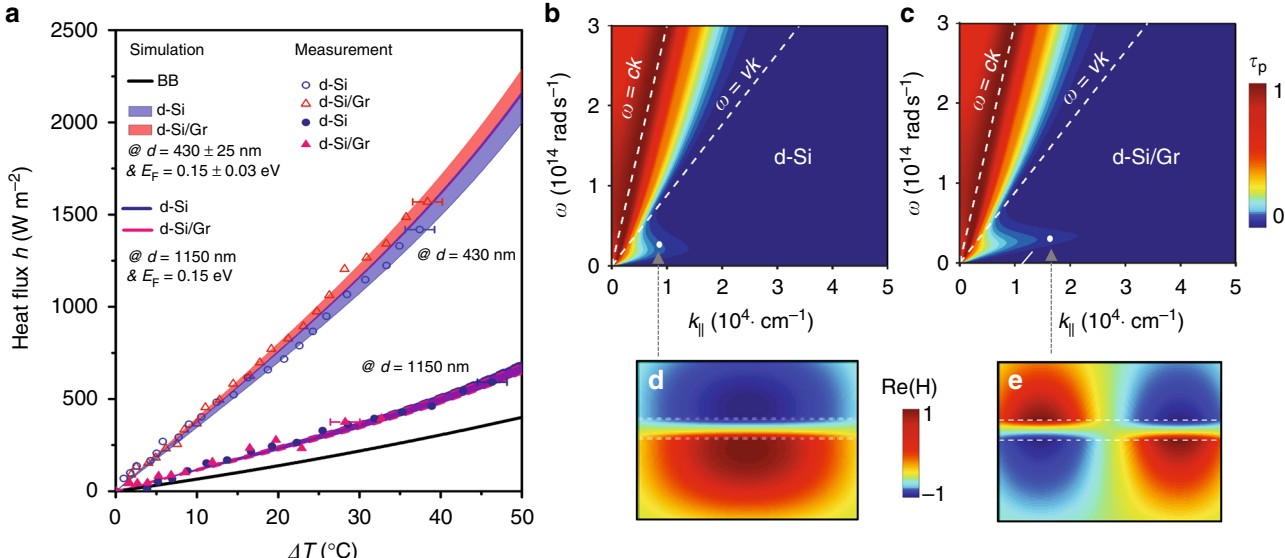

**Fig. 5** Heat flux density and transmission possibility between d-Si/Gr pair. **a** The measured heat flux density (symbols) compared with the analytical predictions (shaded regions) at different temperature difference ($T_r$ is fixed at 30 °C). The solid circles and triangles represent the measured heat flux density between a pair of bare d-Si and d-Si/Gr, respectively, at the gap $d = 1150$ nm. These results can be well reproduced by the theory (solid royal and pink lines) using the measured material parameters. The empty circles and triangles represent the heat flux density measured at the gap of $d = 430$ nm for bare d-Si and d-Si/Gr pairs, respectively. The corresponding blue/red shaded region profiles the theoretical flux density distribution when considering the parametric fluctuations: $E_F = 0.15 \pm 0.03$ eV and $d = 430 \pm 25$ nm. The black line denotes the flux density of Blackbody radiation. The temperature error bar is 2 °C derived from the five times repeating measurement. **b**, **c** p-polarization transmission possibility $\tau_p(\omega, k_\parallel)$ between bare d-Si pair and d-Si/Gr pair, respectively. **d**, **e** The inter-layer mode patterns using magnetic component between bare d-Si pair and d-Si/Gr pair at ($1.23 \times 10^4$ rad cm$^{-1}$, $3.02 \times 10^{13}$ rad s$^{-1}$) and ($1.94 \times 10^4$ rad cm$^{-1}$, $3.02 \times 10^{13}$ rad s$^{-1}$), respectively. Figure 5 by Yang J et al.

measurement accuracy and good quality sample. The former is achieved by our custom-made setup as evidenced by the high agreement of the measured and calculated heat flux densities between the bare d-Si substrates (Fig. 5a). In the experiment, we have taken into account the influence of all the possible background heat loss, in particular including the heat conducted along the electrode wires which previously was not mentioned. For the sample quality, the CVD method is believed able to produce large-scale and high-quality graphene sheets. The transfer process needs to be expertly realized to avoid further damage or contamination. For a gap-size dependent heat flux characterization, the microsphere-plate measurement scheme will be more executable, e.g., by coating a SiO$_2$ microsphere with a graphene micro-disk.

It needs to be pointed out that silicon may not be the best choice to host grapheme if one wants to maximize the heat flux and for this aim polar materials such as silica or BN may be used where the occurrence of plasmon–phonon hybridization can further strengthen the tunneling of evanescent waves[20,45]. However, the Si/Gr heterostructure proposed here can provide a promising framework to harvest thermal energy. A normal TPV cell needs layers of infrared semiconductors (such as InSb or InAs) with an interface p–n junction in order to excite and separate electron–hole charges[52]. From the material perspective, the Gr–Si heterostructure with the existence of Schottky junction could be more realistic compared with the p–n junction which for far-IR optics is not readily available yet. The tunability of Fermi level of graphene will also help to modulate the barrier height and improve the utilization of thermal photons, which becomes critical when the working temperature is relatively low. Direct integration with semiconductors is also advantageous for other device applications such as thermal rectification[53] or thermoelectric sensor[54,55]. In addition, high level chemical doping using species like HNO$_3$ may be employed to further tune the Fermi

level of graphene with larger modulation depth[56] and the number of graphene layers may be optimized to strengthen the near-field coupling and further enhance the heat flux. These methods need more and deep investigations in future.

**Si/Gr-based near-field TPV cell**. In the following, we give a theoretical picture about the performance of the Si/Gr heterostructure based near-field TPV cell. As shown in Fig. 6a, the emitter and the cell (receiver) have the same structural configuration with temperatures of $T_1$ and $T_2$ (=30 °C), respectively. The TPV cell is specifically biased to have a potential difference $V_b$ in order to maximize the output power. A typical interface band diagram for the graphene-covered n-type silicon is drawn in Fig. 6b. Here, $W_{Gr}$ is the work function of graphene and $\Phi_b$ is the energy height of the Schottky barrier. $\chi$ is the electron affinity in silicon. $E_c$ ($E_v$) and $E_F^{Si}$ are the edge of the conduction (valence) band and Fermi level of silicon, respectively. $\Phi_b$ ($= W_{Gr} - \chi$) is dependent on the Fermi level of graphene and can be tuned by charge doping through the bias voltage. In our discussion, we assume a reasonable barrier value $\Phi_b = 0.15$ eV. It means thermal photons with energy larger than this threshold absorbed by the graphene layer are able to cross the barrier and contribute to the photocurrent as indicated in Fig. 6c by the emission of the photons in the shaded region where $\hbar\omega > \hbar\omega_0 \equiv \Phi_b$. Here, the radiation power density spectra of both emitter and receiver are calculated at $T_1 = 700$ °C and $d = 50$ nm. The discontinuity of the curve for the cell side is caused by the excitation of non-thermal photons arising from the potential difference[57], with more details addressed in the Methods section. Under the current device parameters, the main part of absorbed thermal photons are attenuated into heat. Smaller gaps or higher temperature will help to increase the efficiency of photon utilization but at the cost of practical challenge[24,25]. In Fig. 6d, we compare the total radiated

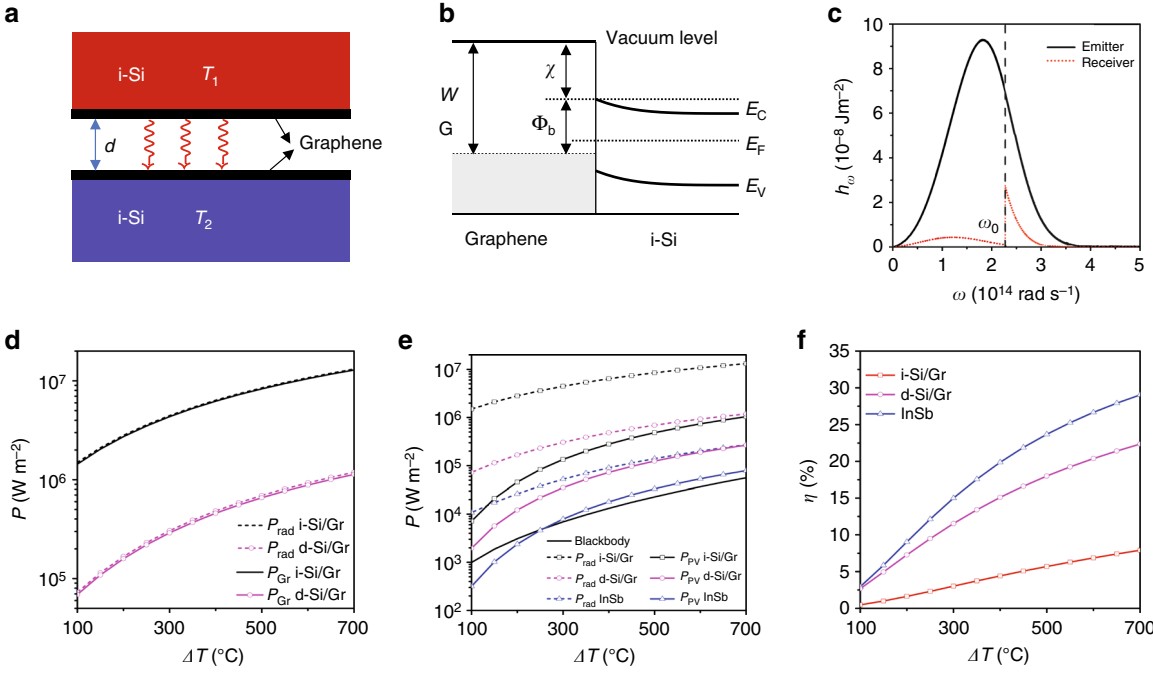

**Fig. 6** Numerical analysis of the Si/Gr-based near-field TPV cell. **a** The schematic of the TPV cell consisting of a pair of Si/Gr heterostructures. The top is emitter with temperature $T_1$ and the bottom is the PV cell (receiver) with temperature $T_2$ (=30 °C). The PV cell is biased to have a potential difference $V_b$. **b** The interface band diagram of the n-Si/Gr Schottky diode for the PV cell. The physical parameters have been defined in the main text. **c** The radiated near-field heat flux spectra of emitter (black) and receiver (red) at 700 and 30 °C, respectively. High energy photons in the shaded region with $\omega > \omega_0 \equiv \Phi_b/\hbar$ could cross the barrier in the cell and contribute to the photocurrent. **d** The radiated power density $P_{rad}$ (dashed line) of the emitter and the portion $P_{Gr}$ (solid line) solely absorbed by the graphene layer on both i-Si (black) and d-Si (pink) substrates at $\Delta T$ ranging from 100 to 700 °C. **e** Power density $P_{PV}$ of the i-Si/Gr (black), d-Si/Gr (pink) and InSb (blue) TPV cells together with their net radiation power $P_{rad}$ (dashed lines). The blackbody radiation power (yellow) is also given for comparison. **f** The corresponding power efficiency $\eta$ (=$P_{PV}/P_{rad}$) for the three cells defined in **e**. The material parameters are specified in the Methods section. The gap is 50 nm. Figure 6 by Yang J et al.

power $P_{rad}$ received by the cell and the portion $P_{Gr}$ absorbed solely by the graphene layer on both insulating and doped silicon substrates as $\Delta T$ ranges from 100 to 700 °C. At this small gap (50 nm), lower loss for the substrate will be more favorable to strengthen the heat flux as the coupling of the graphene layers will be stronger. The thermal photons from the emitter are dominantly absorbed by the graphene layer, which is nearly affected by the loss degree of the substrate. In Fig. 6c, d, we plot the electric power density $P_{PV}$ and efficiency $\eta$ ( = $P_{PV}/P_{rad}$) of two Si/Gr TPV cells at different silicon doping. As a comparison, we also give the result of an InSb (band gap = 0.17 eV[57]) based semiconductor p–n junction at the same gap (50 nm). For both cases, we assume an ideal photocurrent responsivity of 100% (i.e., every absorbed photon satisfying $\hbar\omega > \Phi_b$ will generate one electron–hole pair) in order to examine the limit of the proposed devices. The power density of the Si–Gr junction can be a few times or even one order larger than the semiconductor dependent on the doping of silicon substrate. On the other side, the semiconductor junction has a larger power efficiency due to its relatively low absorption of undesired thermal photons at $\hbar\omega < \Phi_b$. However, for the Si–Gr junction, the power efficiency could be traded off with the cell power density by tuning the substrate doping degree. The results show that in the ideal case the Si–Gr junction has the cost and power density advantages competed with the conventional semiconductor scheme for TPV cells.

## Discussion

In this work, we have given a direct measurement about the graphene plasmon polariton enhanced near-field heat transfer. This effect is explicitly exhibited by the usage of intrinsic silicon

hosting substrates which have ultra-low IR dielectric loss. Super-Planckian thermal radiation efficiency is obtained at a submicron gap between two macroscopic semi-infinite graphene surfaces. We also observed that the charging doping for graphene when highly doped silicon is used as the hosting substrate. In this case, the plasmonic mode coupling of metallic graphene and silicon layers (for silicon, only the surface layer in a penetration depth is effective for the heat transfer) will further enhance the evanescent wave tunneling and lead to a larger heat flux density. The unique property of the Si/Gr heterostructure may provide a promising platform to explore the applications of near-field heat transfer systems for various purposes, in particular for TPVs.

## Methods

**Sample preparation**. The double-side polished Si wafers are used and cut into 2×2 cm pieces of substrates before transfer graphene. They are cleaned by ultrasonic waves sequentially in acetone, methyl alcohol, and isopropyl alcohol. Each step is for 5 min. After these, a 200 nm titanium film is evaporated onto the backside surface of the Si substrates, which are followed by a 500 °C post-annealing for 5 h to release the stress that helps to reduce the bow of substrates. We utilize the mature technique called Trivial Transfer Graphene (TTG)[58] to prepare the i-Si/Gr and d-Si/Gr heterostructures in the following steps. The graphene sheet is first peeled off from the polymer holding substrate and floats above the DI water for more than 2 h before transfer. Silicon substrates are handed using a tweezer to pick up the floating graphene. Great carefulness should be paid to avoiding air bubbles or chemical contamination for Gr sheets. After this, the sample is dried naturally for more than 30 min and then baked for 20 min at 100 °C. Finally, we rinse the sample in a 45 °C acetone for 30 min and then dry it in a 50 °C oven for 10 min. On the surface of the Si/Gr heterostructure, the photoresist (AZ5214E, Resemi, China) is spin coated at 4000 rpm for 29 s and then baked for 6 min at 90 °C. It will lead to an about 1150 nm photoresist layer. A standard UV lithography procedure is employed to fabricate the four supporting posts. The ion coupled plasma (ICP) etching has been used to thin and control the post height for samples with small gaps.

**Heat flux measurement**. For the temperature measurement, the Platinum Resistance Temperature Detector (M222 class A, Heraeus, Germany) is used to monitor the temperature of the emitter and the receiver. We use a source meter (Keithley 2450, Keithley Instruments, USA) to measure the resistance of thermal sensor. The heater is powered by a DC power source (Keithley 2260A-80-27, Keithley Instruments, USA). Its voltage is slowly increased from 0.5 to 3.0 V at a time interval of 15 min between two successive steps, corresponding the total input power changing from 0.05 to 1.05 W. In the process, the receiver is maintained at a constant temperature of 30 °C with the help of TEC. For our firmly clamped near-field system and the usage of conductive grease layer at the boundaries, it is reasonably assumed that the measured copper temperatures are equal to the Si/Gr emitter and receiver[23]. Using the setup described in Fig. 1c, we can accurately estimate the background heat loss of the measurement platform. Since the total input is known, the net heat conduction through the vacuum gap could be obtained accurately. The details have been described in the main text.

**Heat flux calculation**. The near-field thermal radiation is calculated using the classic fluctuating electrodynamics theory[23,41,59]. For the Si pair without graphene, the near-field radiative heat flux is calculated by:

$$h(T_1, T_2, d) = \int_0^\infty \frac{d\omega}{4\pi^2} [\Theta(\omega, T_1) - \Theta(\omega, T_2)]$$
$$\int_0^\infty dk\, k \left[ \tau_s(\omega, k) + \tau_p(\omega, k) \right] \tag{1}$$

where $\Theta(\omega, T) = \hbar\omega / \{\exp[\hbar\omega/(k_B T)] - 1\}$ is the power density of Planck's blackbody radiation and $\tau_{s,p}$ is the transmission probability of the p-polarizations and s-polarizations, derived as

$$\tau_{\alpha=s,p} = \begin{cases} \frac{(1 - |r_\alpha^{01}|^2)(1 - |r_\alpha^{02}|)}{|1 - \exp(2ik_0^z d) r_\alpha^{01} r_\alpha^{02}|^2}, & k_0^z \le k_0 \\ \frac{4\Im(r_\alpha^{01})\Im(r_\alpha^{02})\exp(-2|k_0^z|d)}{|1 - \exp(-2|k_0^z|d) r_\alpha^{01} r_\alpha^{02}|^2}, & k_0^z > k_0 \end{cases} \tag{2}$$

where $r_\alpha^{ij}$ is the Fresnel reflection coefficient of the interface. In the calculation, $i = 0$ represents the vacuum gap and $j = 1$ or 2 denotes the emitter or receiver. When a monolayer graphene is introduced on silicon, the Fresnel reflection coefficient in Eq. (2) will change into[59–61]:

$$r_s^{01} = \frac{k_{0z} - k_{1z} - \delta\mu_0\omega}{k_{0z} + k_{1z} + \delta\mu_0\omega}$$
$$r_p^{01} = \frac{\varepsilon_1 k_{0z} - k_{1z} + \frac{\delta k_{0z} k_{1z}}{\omega\varepsilon_0}}{\varepsilon_1 k_{0z} + k_{1z} + \frac{\delta k_{0z} k_{1z}}{\omega\varepsilon_0}} \tag{3}$$

where $\delta$ means the sheet conductivity of graphene and $k_{i,z}$ ($i = 0, 1$) denotes the normal component of wave vector in vacuum or dielectric, $\varepsilon_i$ denotes the electric permittivity and $\mu_0$ stands the permeability of vacuum.

**Material parameters evaluation**. The doped silicon has a tunable plasma frequency dependent on the carrier concentration. Typically, it can be described by the Drude model given below[14]

$$\varepsilon(\omega) = \varepsilon_\infty - \frac{\omega_{p,h}^2}{\omega^2 + i\gamma_h\omega} - \frac{\omega_{p,e}^2}{\omega^2 + i\gamma_e\omega} \tag{4}$$

where $\varepsilon_\infty$ (=11.70 for Si) is the dielectric constant at the infinite frequency, $\omega_{p,h}$ ($\omega_{p,e}$) and $\gamma_h$ ($\gamma_e$) represent the plasma and collision frequencies for free carriers of hole (electron), respectively. We measure the reflectance of the doped silicon by FTIR at a 13° incident angle. Such a small incident angle will give a small coefficient change compared with those at the normal incidence. The measured reflectance spectra at different sample temperature are plotted in Fig. 5a, which show no obvious difference. From the curve fitting based on the Drude model given in Eq. (4), we obtain the parametric data of our doped silicon: $\varepsilon_\infty = 11.67$, $\omega_{p,h} = 3.9 \times 10^8$ rad s$^{-1}$, $\gamma_h = 4.1 \times 10^{13}$ rad s$^{-1}$, $\omega_{p,e} = 5.6 \times 10^{14}$ rad s$^{-1}$, and $\gamma_e = 1.1 \times 10^{14}$ rad s$^{-1}$. Based on these input, the permittivity spectra of the doped silicon are plotted in Fig. 5b. When the dielectric properties of silicon are known, we can numerically obtain the real and imaginary sheet impedance of the graphene monolayer from the measured reflectance of the heterostructure as represented by the blue and red squares in Fig. 5c. Then a Drude model for graphene conductivity is applied to fitting the measured curves

$$\sigma_{intra}(\omega) = \frac{ie^2 |E_F|}{\pi\hbar^2(\omega + i\tau_{Gr}^{-1})} \tag{5}$$

which gives the Fermi level of graphene at $E_F = 0.15 \pm 0.03$ eV.

**Power density calculation of the TPV cell**. For the PV cell, the potential difference $V_b$ (=$\Phi_b/e$) will impose a chemical potential $\mu_c$ on the emitted photons

with energy larger than the barrier and in this case, the PV cell will give a backward radiation power to the emitter by[57]:

$$P_{cell}(T_2, d) = \int_0^\infty \frac{d\omega}{4\pi^2} \frac{\hbar\omega}{\exp[(\hbar\omega - \mu_c)/(k_B T)] - 1}$$
$$\int_0^\infty dk\, k \left[ \tau_p(\omega, k) + \tau_s(\omega, k) \right] \tag{6}$$

where $\mu_c = \Phi_b$ for $\omega > \omega_0$ and 0 for $\omega < \omega_0$. The emitter's radiation still follows Eq. (1). Then from their difference, we could obtain the net radiation power $P_{rad}$ of the Si/Gr cell system. The power $P_{Gr}$ absorbed by the graphene layer is calculated by subtracting $P_{rad}$ with the heat flux passing through the bottom surface of graphene. When we assume every photon with frequency $\omega > \omega_0$ absorbed by the graphene layer could cross the barrier and excite an electron, the photocurrent $I_{ph}$ generated in the cell is simply calculated by multiplying the electron charge $e$ with the number of these photons. Regarding the voltage of the cell, it has a limit defined by $V_b = (1 - T_2/T_1)\Phi_b/e$. In our calculation of Fig. 6e, we take a slight smaller value to have the maximum cell power $P_{PV} = I_{ph} \times V_b$. The power efficiency of the cell is defined by $P_{PV}/P_{rad}$. In our calculation, the parameters for i-Si are same with those used in Fig. 3 and for d-Si are $\varepsilon_\infty = 11.7$, $\omega_{p,h} = 8.08 \times 10^{14}$ rad s$^{-1}$ and $\gamma_e = 9.34 \times 10^{13}$ rad s$^{-1}$. In the two cases, $E_F = 0.5$ eV and $\tau_{Gr} = 100$ fs for Gr. The parameters for InSb are obtained from ref. [62].

## Data availability

The data that support the findings of this study are available from the corresponding author upon request.

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

## Acknowledgements

We are grateful to the partial supports from the National Key Research and Development Program of China (No. 2017YFA0205700), NSFC (61775195) and NSFC of Zhejiang Province (LR15F050001& LZ17A040001).

## Author contributions

J.Y. conducted the most of the experiment and theoretical work. Y.S. participated in the theoretical calculation. Y.F., S.G., and W.D. participated in the measurement. S.H. participated in the discussion on TPV cell. Y.M. supervised the work. All the authors participated in the writing of the manuscript.

## Additional information

**Competing interests:** The authors declare no competing interests.

