## [Peer Review File · Nature Communications]

Reviewers' comments:

Reviewer #1 (Remarks to the Author):

In this work, the authors measured the plasmon-mediated thermal radiation between two graphene sheets seated on silicon substrates. They observed a super-Planckian radiation with efficiency 4.5 times larger than the blackbody limit at a 430-nm vacuum gap by using intrinsic insulating silicon. Charging doping for graphene with highly doped silicon can further enhance the evanescent wave tunneling, leading to a larger heat flux density. Such graphene/silicon Schottky structure for plasmon-mediated thermal radiation has been proposed in previous works, such as Ref. 25, Ref. 30, and Ref. 59 in the manuscript. Therefore, the main novelty of this manuscript is the demonstration of a direct experimental measurement on the plasmon-mediated thermal radiation effect. This work should be of interest to the research community. I recommend the publication of this manuscript after addressing the following questions.

- (1) Monolayer graphene was adopted in this work. As we know, thickness of graphene is an important factor that controls the absorption. How does the graphene layer number (2-3 layers or even more by repeating the transfer process) influence the experimental results?
- (2) Instead of graphene doping by using highly doped silicon wafer, have the authors tried to dope the graphene layer with other chemical species, such as HNO₃ etc.? Since these chemical species can offer higher doping level than doped silicon.
- (3) The authors claimed that the graphene/silicon Schottky junction can have application in thermophotovoltaic (TPV) cells. However, can the authors provide any experimental evidence on this kind of cell?

Reviewer #3 (Remarks to the Author):

Authors have done a careful and innovative measurement of nanoscale heat transfer through a vacuum gap with a distance down to 430 nm and lateral dimensions of 2 cm by 2 cm for graphene covered silicon substrate. The effects of graphene on intrinsic and doped silicon have been studied and the experiments seem to agree with the modeling. The innovative aspects are the use of magnetic force to hold the samples together, the use of photoresist pads to support the samples and create the vacuum gap, and the study of graphene covered materials. Near-field radiation may have important applications in photovoltaic generation, as also envisioned by the authors in this paper.

A few questions may be raised and can be explained more clearly.

1. Please provide more details on the gap thickness determination. In Fig. 2d, the interference effect is NOT due to 430 nm vacuum gap. With a 430 nm gap, there won't be such oscillations, which correspond to a thickness of an order of magnitude larger!!
2. Will the weight of the structure shown in Fig. 1a affect the force provided to the gap? If so, then the transmittance measurement is for 90 deg orientation, right?
3. On page 3, the Si wafer has "a roughness about 1-2 nm and a maximum bow value less than 20 nm." How is the bow value determined and for how large an area?
4. How large is P_{AZ} as compared with P_r ?
5. Could you comment on how to control the thickness and uniformity of the thickness of the photoresist?

6. Page 10, Heat flux measurement: sentence 1 says you used a resistance thermometer; sentence 2 says you measured the resistance of thermocouples? Is "thermocouples" a typo or did you actually use a thermocouple? Then you should measure its voltage not resistance!

7. There are some other terminology issues that the authors may want to compare with standard literature to further improve.

To Reviewer1:

Comment: In this work, the authors measured the plasmon-mediated thermal radiation between two graphene sheets seated on silicon substrates. They observed a super-Planckian radiation with efficiency 4.5 times larger than the blackbody limit at a 430-nm vacuum gap by using intrinsic insulating silicon. Charging doping for graphene with highly doped silicon can further enhance the evanescent wave tunneling, leading to a larger heat flux density. Such graphene/silicon Schottky structure for plasmon-mediated thermal radiation has been proposed in previous works, such as Ref. 25, Ref. 30, and Ref. 59 in the manuscript. Therefore, the main novelty of this manuscript is the demonstration of a direct experimental measurement on the plasmon-mediated thermal radiation effect. This work should be of interest to the research community. I recommend the publication of this manuscript after addressing the following questions.

Answer: Thank you very much for your positive comments!

Q1. Monolayer graphene was adopted in this work. As we known, thickness of graphene is an important factor that controls the absorption. How does the graphene layer number (2-3 layers or even more by repeating the transfer process) influence the experimental results?

Answer: Yes, the number of the Gr-layer is a critical factor that will affect the heat transfer efficiency. As shown in Fig. R1(a) below, the heat flux becomes much larger for the case of double or triple Gr layers compared with the single layer case as explored in the manuscript. However, in the near-field system, it doesn't mean that a sample with more Gr layers is always better because this effect is affected not only by material loss but also more importantly by the near-field coupling strength, which is different from the far-field absorption. The number of layers and their spacing will affect the near-field coupling and thus the heat transfer efficiency. As shown here, the sample with two graphene layers has the highest heat flux, which is more obvious as shown in Fig. R1(b) when the distance between the receiver and the emitter gets smaller. From the transmission possibility maps plotted in Figs. R1(d)-R1(f), it is seen that the band from the plasmonic coupling is extended both in the time and spatial frequency domains when the number of the Gr layer increases. However, it splits and becomes more complex for the sample having triple graphene layers due to the interlayer coupling, which in turn complicates the field coupling between the emitter and the receiver and affects the overall heat transfer efficiency. We haven't experimentally examined these results yet. The main difficulty is the precise controlling of the gap distance between the neighboring Gr sheet in macroscopic scales. As shown in Fig. R1(c), the heat flux is very sensitive to the gap size. On the other hand, the substrate covered with a single layer graphene has a roughness about 50 nm (shown by the AFM picture in Fig. 2a in the main text). It is hard to control and predict the exact spacing when a second or more layers of graphene is

transferred. We will seek the experimental possibility in future by using a bilayer graphene although in this case, the Fermi level and other optical parameters may change.

Fig R1 Influence of the number of Gr layer. (a) Heat flux as a function of temperature difference for 1-layer (black), 2-layers (red) and 3-layers (blue) Gr samples. We assume there is a 1-nm gap between two neighboring Gr sublayers. (b) Heat flux as a function of the distance between the receiver and the emitter. (c) Heat flux as a function of the spacing of the neighboring Gr-layers for the sample of double Gr layers. Three different Fermi levels for Gr are considered here. In (a, c), the distance between the emitter and receiver is fixed at 500 nm. (d-f) Transmission possibility of the single, double and triple Gr-layer systems at $d = 500$ nm, respectively. The insets are the magnetic field patterns from left to right at $\omega = 2.5 \times 10^{13}$ rad s⁻¹ and $k_{||} = 2.3 \times 10^4$ rad cm⁻¹, $\omega = 6.3 \times 10^{13}$ rad s⁻¹ and $k_{||} = 1.5 \times 10^4$ rad cm⁻¹, $\omega = 6.3 \times 10^{13}$ rad s⁻¹ and $k_{||} = 1.7 \times 10^4$ rad cm⁻¹.

Our corresponding modifications:

- 1) Lines 305-307, page 8, we added the sentences “the number of graphene layers may be optimized to strengthen the near-field coupling and further enhance the heat flux. These methods need more and deep investigations in future.”.

Q2. Instead of graphene doping by using highly doped silicon wafer, have the authors tried to dope the graphene layer with other chemical species, such as HNO₃ etc.? Since these chemical species can offer higher doping level than doped silicon.

Answer: Yes, chemical doping could raise the Fermi level much higher. It could be a good suggestion if the morphology of the sample can be well controlled, too. Currently, we don’t have the experience to dope graphene using different species but would like to try it in the near future through collaborating with others. The current work is mainly to verify that the infrared plasmon polariton of graphene can play a key role in manipulating and enhancing the thermal emission in

the near-field scenario.

Our corresponding modifications:

- 1) Lines 304-305, page 8, we added the description “... high level chemical doping using species like HNO_3 may be employed to further tune the Fermi level of graphene with larger modulation depth⁵⁶...”.
- 2) A new reference (Ref. 56) has been added to explain the high-level doping effect.

Q3. The authors claimed that the graphene/silicon Schottky junction can have application in thermophotovoltaic (TPV) cells. However, can the authors provide any experiments evidence on this kind of cell?

Answer: In the present stage, we cannot provide the experiment evidence about the near-field TPV cell mainly due to the technical challenges. Experimentally, it will require a large temperature difference (> 300 K) at a smaller gap (< 200 nm). The current measurement scheme cannot realize these conditions. But we are developing a new methodology now to satisfy these requirements by fully taking the advantages of magnetic attracting force. Quite recently, researchers from another group just gave the first experimental verification about the near-field TPV cell using narrow band-gap semiconductors in micron scales [Ref.: A. Fiorino, et al, Nanogap near-field thermophotovoltaics, Nature Nanotechnology, published online on 16 July 2018, <https://doi.org/10.1038/s41565-018-0172-5>], which shows the technical difficulties related with this application could be overcome with advanced designs.

To Reviewer 2:

Comment: Authors have done a careful and innovative measurements of nanoscale heat transfer through a vacuum gap with a distance down to 430 nm and lateral dimensions of 2 cm by 2 cm for graphene covered silicon substrate. The effects of graphene on intrinsic and doped silicon have been studied and the experiments seem to agree with the modeling. The innovative aspects are the use of magnetic force to hold the samples together, the use of photoresist pads to support the samples and create the vacuum gap, and the study of graphene covered materials. Near-field radiation may have important applications in photovoltaic generation, as also envisioned by the authors in this paper.

Answer: The authors sincerely appreciate the positive comments!

A few questions may be raised and can be explained more clear.

Q1: Please provide more details on the gap thickness determination. In Fig. 2d, the interference effect is NOT due to 430 nm vacuum gap. With a 430 nm gap, there won't not be such oscillations, which corresponding to a thickness of an order of magnitude larger!!

Answer: Thanks for pointing it out and we are truly sorry for the mistake. The FTIR curve in the original Fig. 2d doesn't correspond to the real sample used to measure the near-field heat flux. There was an information exchange mistake among us. The experiment really took a very long time. Most of it was spent on the preparation of the sample. The gap distance is mainly determined by the height of the supporting photoresist posts. But it has a deviation/fluctuation for the real

sample due to some inevitable issues. To evaluate this, in the beginning of our experiment, we prepared several samples consisting of bare silicon substrates that had different gap distances. As shown in Fig. R2 below, the measured gap value has a good linear correspondence with the target when the gap is within few micron meters. It is determined by fitting the experimental FTIR curves collected at various spatial points by theoretical transmission spectra calculated using different gap values. For this, the measured optical parameters for silicon are used. As shown in the right panel of Fig. R2 below, the blue and red theoretical lines basically enclose the measured curves, from which we estimated the distribution of the gap in a sample.

Fig. R2 Determination of the gap distance. The left figure plots the measured gap distance vs the target value for Si-Si pair samples. The right side gives the FTIR curves measured (dashed lines) and calculated (solid lines) at three example target distances: $d_{\text{target}} = 3.7, 1.4$ and $0.43 \mu\text{m}$, respectively. The blue and red solid lines define the edges of the experimental data.

Our corresponding modifications:

- 1) Fig. 2 with its caption has been updated by the following one:

Figure 2 Characterization of i-Si/Gr properties. (a) A typical 3D AFM picture of the i-Si/Gr sample. The largest roughness of the sample is about 50 nm. (b) The Raman spectrum of the i-Si/Gr sample. The monolayer feature of graphene is identified by the strong 2D resonance peak. (c) The absorption spectrum of graphene on intrinsic silicon. The red dots are measured results by FTIR and the blue line is a Gaussian function fitting. T_{Gr} and T_{Si} denote the transmittance with and without the graphene cover, respectively. The “ $2E_F$ onset” of the inter-band transition is estimated at about 4832.8 cm^{-1} , corresponding a Fermi level at $E_F = 0.27 \text{ eV}$. (d) Measured (dashed lines) and calculated (solid lines) FTIR spectra at three different target gap sizes of 3.7, 1.4 and 0.43 μm . The uncertainty of gap size is considered here so that the calculated lines (red and blue) could cover the experimental curves measured at various spatial points. The statistic value for the smallest gap we designed is about $430 \pm 25 \text{ nm}$. (e) Correspondence between the measured and the target gap sizes. The gap uncertainty is shown by the error bar. The black line is a guidance for eye, which shows nearly a linear relationship when the gap value is within few micron meters.

2) Lines 168-175, page 5, the original sentences have been replaced by the words “...The gap distance d of the silicon substrates is mainly decided by the height of the photoresist posts. The real value may have a certain distribution due to the bow of wafer or roughness, etc. As shown in Fig. 2d, we use the measured FTIR spectra (dashed lines) and the theoretical ones (solid lines) to evaluate the gap value. The red and blue lines calculated at different d represent the edges of the experimental lines collected at various spatial points. Fig. 2e shows there is a good linear correspondence between the measured and the target gap distances when it is within few micron meters. The uncertainty is about 25 nm at a central distance of 430 nm and gets larger when the distance increases...”.

Q2. Will the weight of the structure shown in Fig. 1a affect the force provided to the gap? If so,

then the transmittance measurement is for 90 deg orientation, right?

Answer: Yes, the weight will affect the force applied to the gap. But in practice, they are balanced by the upward bending force caused by the four elastic metal wires (belonging to the electrodes and temperature sensors; behaving like four legs for the top emitter). That's the reason we applied an external magnetic force to make sure the good physical contact between the upper and the lower part of the measurement structure. Our FTIR apparatus setup allows us to do both measurements, i.e., 0 and 90 deg orientation. Their difference is very small when the weight and elastic forces are well controlled.

Q3. On page 3, the Si wafer has "a roughness about 1-2 nm and a maximum bow value less than 20 nm." How is the bow value determined and for how large an area?

Answer: The bow value is determined by a Laser interferometer (ZYGO OMP-035/M) with an area of 2cm× 2 cm, as shown below by Fig. R3. It is widely used to inspect the surface geometries of optical instruments like lens.

Fig R3 Surface profile of our 2 cm×2 cm silicon substrate. The color represents the profile variations of the wafer along the z-axis measured by a Laser interferometer (ZYGO OMP-035/M).

Our corresponding modifications:

- 1) Line 99, page 3, we added the word "...determined by a Laser interferometer (ZYGO OMP-035/M) with an area of 2cm× 2 cm..."

Q4. How large is P_{AZ} as compared with P_r ?

Answer: In terms of our setup and used materials parameters, P_r is about four times larger than P_{AZ} .

Our corresponding modifications:

- 1) Lines 190-192, page 5, we have added the sentence "...The total radiation heat flow at this case is about 0.48 W for our sample, which is nearly four times larger than that ($P_{AZ} \sim 0.12$ W) conducted along the four supporting posts..."

Q5. Could you comment on how to control the thickness and uniformity of the thickness of the photoresist?

Answer: First we prepare the posts with relatively large height controlled by the rotation speed of spin coating and solution concentration. Then we use standard ICP dry etching to thin the post to obtain the desired height (i.e., target gap distance) by controlling the etching time. In the process, we have used an alpha-step surface profiler to monitor the height and evaluate the uniformity. In addition, the quality of each post could be controlled individually through a mask.

Q6. Page 10, Heat flux measurement: sentence 1 says you used a resistance thermometer; sentence 2 says you measured the resistance of thermocouples? Is "thermocouples" a typo or did you actually use a thermocouple? Then you should measure its voltage not resistance!

Answer: Thanks for pointing out the error. Yes, we used the resistance measurement to determine the temperature. It has been corrected.

Q7. There are some other terminology issue that the authors may want to compare with standard literature to further improve.

Answer: Thanks for the question. The expressions have been improved all over the paper.

Best regards,
Yungui MA,
on behalf of all the authors

REVIEWERS' COMMENTS:

Reviewer #1 (Remarks to the Author):

The authors have addressed most of the reviewers' concerns in the revised manuscript. Considering that this manuscript is the first experimental verification of plasmon-mediated thermal radiation between Gr-sheets seated on silicon, I recommend the acceptance of this manuscript as it is.

Reviewer #3 (Remarks to the Author):

Fig. 2 has been corrected and I do not have further comments.

To Reviewer #1 (Remarks to the Author):

The authors have addressed most of the reviewers' concerns in the revised manuscript. Considering that this manuscript is the first experimental verification of plasmon-mediated thermal radiation between Gr-sheets seated on silicon, I recommend the acceptance of this manuscript as it is.

Response: Thank you very much!

To Reviewer #3 (Remarks to the Author):

Fig. 2 has been corrected and I do not have further comment.

Response: Thank you very much!

Best regards,
Yungui MA,
on behalf of all the authors